# Genome-Wide Identification and Characterization of RNA/DNA Differences Associated with Drought Response in Wheat

**DOI:** 10.3390/ijms23031405

**Published:** 2022-01-26

**Authors:** Yan Pan, Mengqi Li, Jiaqian Huang, Wenqiu Pan, Tingrui Shi, Qifan Guo, Guang Yang, Xiaojun Nie

**Affiliations:** 1State Key Laboratory of Crop Stress Biology in Arid Areas, College of Agronomy, Northwest A&F University, Yangling 712100, China; py81212256@163.com (Y.P.); lmq13292903600@163.com (M.L.); Hjqian1998@163.com (J.H.); wenqiu_pan@nwafu.edu.cn (W.P.); shitingrui@nwafu.edu.cn (T.S.); gqf520521@163.com (Q.G.); drbiology@aliyuan.com (G.Y.); 2ICARDA-NWSUAF Joint Research Center, Yangling 712100, China

**Keywords:** drought, miRNA targeting, RNA secondary structure, RNA/DNA difference, wheat

## Abstract

RNA/DNA difference (RDD) is a post-transcriptional RNA modification to enrich genetic information, widely involved in regulating diverse biological processes in eukaryotes. RDDs in the wheat nuclear genome, especially those associated with drought response or tolerance, were not well studied up to now. In this study, we investigated the RDDs related to drought response based on the RNA-seq data of drought-stressed and control samples in wheat. In total, 21,782 unique RDDs were identified, of which 265 were found to be drought-induced, representing the first drought-responsive RDD landscape in the wheat nuclear genome. The drought-responsive RDDs were located in 69 genes, of which 35 were differentially expressed under drought stress. Furthermore, the effects of RNA/DNA differences were investigated, showing that they could result in changes of RNA secondary structure, miRNA-target binding as well as protein conserved domains in the RDD-containing genes. In particular, the A to C mutation in TraesCS2A02G053100 (orthology to *OsRLCK*) led to the loss of tae-miR9657b-5p targeting, indicating that RNA/DNA difference might mediate miRNA to regulate the drought-response process. This study reported the first drought-responsive RDDs in the wheat nuclear genome. It sheds light on the roles of RDD in drought tolerance, and may also contribute to wheat genetic improvement based on epi-transcriptome methods.

## 1. Introduction

Wheat (*Triticum aestivum* L.) is one of the most important cereal crops all over the world, providing the staple food source for about 30% of the global population and occupying approximately 20% of the world’s cultivated lands [1]. Continuity of wheat production holds promise for ensuring global food security despite the challenges of population growth and global climate change [2,3,4]. It is estimated that wheat production should increase by 1.5% annually to meet the requirements of increasing population [5]. Drought is one of the most destructive environmental stresses that impair plant growth and development, reducing yields [6]. Recently, drought has become the most serious constraint to wheat production, causing about 5.5% yield loss every year [7,8]. Therefore, better understanding of the molecular mechanisms of drought response in wheat is of great significance for genetic improvement and breeding of drought-tolerant varieties.

RNA/DNA differences (RDDs) are phenomena of base insertion, deletion or modification, occurring when DNA is transcribed into RNA [9]. This widespread post-transcriptional modification mechanism contributes significantly to the diversity and plasticity of cellular RNA signatures, increasing proteomic diversity by modifying the sequence of primary transcripts that do not operate completely or partially [10,11]. Together with alternative splicing, RDD provides a crucial way to enrich genetic information and diversify transcripts, playing an important role in regulating growth, development and stress response in eukaryotes [12]. ‘Substitution’ by simple base modification is the most common type of RDD, and is widely identified in plant organelle and higher eukaryotic nuclei genomes [13,14] as well as some viral sequences [15]. In mammals, the common type of RDD is A-to-I (G) (adenosine to inosine, guanosine), which is mainly catalyzed by double-stranded RNA-specific ADAR family proteins [16]. It is also called as A-to-G editing because the inosine in RNA is interpreted as guanosine (G) by the translation mechanism [17]. At the same time, A-to-I transformation independent of ADAR enzyme was also found in fungi [16]. However, ADAR-like enzymes have not been found in plants [18]. Another type, cytidine to uridine (C-to-U), is found to be catalyzed by APOBECs activation in humans [19], while the mechanism of the other remaining 10 modification types is still unclear [20,21]. In plants, RNA/DNA difference is generally found in organellar transcripts, and mainly regulated by the specific pentatripeptide repeat proteins encoded by the nuclear genome that deaminate cytidine to uridine [11,22,23]. There are two main types of PPR proteins: P and PLS [24]. The PLS group seems to be the most relevant class for RNA editing in plants with a conserved extended domain at the C-terminal and DYW motif [25,26,27]. The DYW motif encodes the active center of cytidine deaminase, which may be responsible for C-to-U modification [25].

High-throughput RNA-seq technology makes it possible to investigate the transcript diversity and variation from the perspective of whole transcriptome level, which provides an efficient, unbiased, economical and comprehensive tool to identify RNA/DNA differences [28,29]. Extensive researches have utilized this method to identify the RNA editome and RNA/DNA difference landscape in yeast, humans and other model species, demonstrating the prevalence and significance of RNA/DNA differences [9]. However, the study of RNA/DNA difference in plants lags behind compared to animals and fungi, especially as genome-wide identification of RNA/DNA difference in plant nuclear genome was only conducted in Arabidopsis [30]. With the completion of a high-quality reference genome and a number of publicly available RNA-seq datasets, genome-wide identification of RNA/DNA differences in wheat nuclear genome become possible. Recently, Yang et al. reported the FHB-responsive RDDs in wheat nuclear genome through the RNA-seq dataset of the Fg-infected and mock-infected spike samples [31]. To obtain some insights on the RDDs related to drought stress, we used the 18 RNA-Seq datasets from 3 tissues under drought and control conditions of the cultivar Chinese spring (CS42) at the seedling stage to identify the drought-responsive RDDs here. Then, the effects of these RDDs on gene expression, RNA secondary structure, miRNA-target binding as well as protein conserved domain of the edited genes were also systematically investigated. This study aims to illuminate whether RDDs is occurred under drought stress and then to identify some drought-responsive RDD event, which will lay the foundation for better understanding the epitranscriptomic mechanism of drought response and tolerance in wheat.

## 2. Results

### 2.1. Identification of RNA/DNA Differences Based on RNA-Seq Data in Wheat

To identify potential RNA/DNA differences, we used the 18 RNA-Seq datasets of 3 tissues collected from the genotype CS42 seedlings under drought and control conditions, which is the same accession used to generate the reference genome. Based on the approach described in the Methods section, a total of 15,339, 10,821 and 12,502 RDDs were found in the control samples of crown, leaf and root respectively, while 15,167, 10,960, 12,429 RDDs were identified in drought-stressed samples of the crown, leaf and root, respectively (Figure 1A,B). It seems that the CK sample displayed slightly higher RDDs compared to the drought-treated sample in all of the 3 tissues. Meanwhile, for both the CK and drought condition, leaves had the lowest RDDs. These results indicated that RDDs were widely found in wheat nuclear genome. Furthermore, 21,782 unique RDDs were obtained through combining these identified RDDs, of which 8675 appeared in protein-coding genes (Appendix A). Among them, 4205 genes had one RDD, followed by 1794, 1068 and 588 genes with 2, 3 and 4 RDDs, and 1020 genes with more than 5 RDDs, respectively. Overall, 15,623, 11,216, 12,809 RDDs corresponding to 6618, 4959 and 5331 genes were found in crown, leaves and root respectively, of which 6645 RDDs in 3151 genes were commonly found in all of the three tissues (Figure 1C). For RNA/DNA difference type, all of the 12 conversion types were found, of which the conversion between C and T accounted for 24.19% of all sites, and conversion between A and G accounted for 24.31% respectively, which were the two canonical RNA/DNA difference types [31]. Additionally, C to G (7.18%) and G to C (7.09%), A to C (6.99%), T to G (6.86%), T to A (6.67%), A to T (6.14%), and C to A (5.40%) as well as G to T (5.17%) were also identified (Figure 1D). Gene annotation showed that 12,325, 2770, 1250 and 604 RDDs were located in the CDS, intron, 5′UTR and 3′UTR regions, respectively. Furthermore, 4513 and 3948 RDDs were annotated as missense variants and synonymous variants, accounting for 20.72% and 18.13% of RDDs, respectively, of which 189 RDDs could result in the premature stop or loss of the start/termination codons (Figure 1E).

### 2.2. Identification of RDDs Associated with Drought Response

Based on the above identified RDDs, we further comprehensively screened the potential RDDs associated with drought stress through comparing the CK and stressed samples combined with IGV viewing. In total, 265 drought-responsive RDDs were obtained, of which 78, 97 and 90 RDDs were found in crown, leaves and root, respectively (Appendix A). Surprisingly, drought-responsive RDD events displayed strong tissue specificity that none was shared by the three tissues. Among them, 76, 100 and 89 RDDs were located on subgenome A, B and D, corresponding to 54, 69 and 53 protein-coding genes, respectively (Figure 2A). Furthermore, 186 and 79 drought-responsive RDDs were located in CDS and UTR regions respectively, of which, 75 RDDs could cause missense variants (Figure 2B). These RDDs that could cause amino acid changes might play an important regulatory role in coping with drought stress response and tolerance. For difference type, there also all 12 types were found. Similar to the overall RDDs, the two canonical types conversion between C and T together with conversion between A and G were also the most abundant types (Figure 2C), similar to wild grapevine [32,33]. It is obvious that base transition events were higher than base transversion in these RDDs, although transversions should have twice higher occurrence frequency (transition/transversion ratio was 1.55), suggesting that these RDDs were not randomly occurring and related to the specific transcription regulation mechanism underlying the RND/DNA differences resulting from drought stress.

Thus, we further investigated the physical position of these drought-responsive RDDs. Results showed that they were mainly located about 10kb upstream or downstream of the translational start site (TSS) of their corresponding genes (Figure 2D); it is suggested that RDDs may impact on the expression of the corresponding genes. Compared to the genes that did not contain RDDs, the genes containing RDDs have significantly longer gene, exon, CDS and intron size (Figure 2E–H). Furthermore, the difference occurrence efficiency was also calculated based on the ratio of the number of difference reads to the total reads for each RDD site (Figure 3A). Results found that these RDDs under the drought condition displayed significantly higher difference efficiency than that of CK. The efficiency value of the drought-responsive RDDs ranged from 0.07 to 0.83, with most showing 0.1 to 0.2 (108 sites), followed by 0.2 to 0.3 (55 sites), 0.3 to 0.4 (41 sites), 0.4 to 0.5 (22 sites), 0.5 to 0.6 (13 sites), 0.6 to 0.7 (10 sites), 0 to 0.1 (9 sites), 0.7 to 0.8 (4 sites) and 0.8 to 0.9 (3 sites). The efficiency density of the three tissues is skewed to the left (Figure 3B), suggesting that all drought-responsive RDDs had relatively low difference efficiency in all tissues. Finally, we randomly selected one site in TraesCS3A02G045300 to perform experimental verification. Total RNA and DNA was extracted from the leaves. Results showed that the T to C mutation was not found in genomic DNA or the cDNA of CK samples, but appeared in the cDNA of drought stressed samples (Figure 3C), indicating that it is an actual drought-responsive RDD site.

### 2.3. Effects of RDDs on Gene Expression

To explore the relationship between RDDs and gene expression, we further investigated the expression level of the genes containing drought-responsive RDDs under CK and drought conditions. In total, 35 unique genes were found to display differential expression patterns (Figure 4A), suggesting they might play a role in the regulation process of drought response. In detail, 11 (6 down-regulated and 5 up-regulated), 15 (9 down-regulated and 6 up-regulated) and 9 (8 down-regulated and 1 up-regulated) differential RDD-containing genes were found in root, leaf and crown, respectively. Among them, TraesCS2B02G038700 is annotated to encode a Chalcone synthase. Chalcone synthase is the key enzyme controlling flavonoid biosynthesis, which has been proven to monitor the drought stress tolerance [34]. TraesCS2B02G079100 is the orthology of *OsRBCS*, which is a drought stress related marker gene [35]. These results indicated that RDDs in these drought related genes might mediate their expression to regulate the drought response process. Furthermore, functional enrichment found that they are significantly enriched in stress response related terms, such as ‘defense response’, ‘response to oxidative stress’ and ‘response to desiccation’. In addition, they are also enriched in terms related to the structure of DNA or RNA, including ‘mRNA binding’, ‘DNA binding’. In KEGG pathway, they are enriched in the biosynthesis of secondary metabolites, flavonoids biosynthesis, and zeatin biosynthesis, all of which were also related to the drought response in plants (Figure 4B) (Appendix A).

### 2.4. Effect of RDDs on RNA Secondary Structure and MiRNA Targeting

Secondary structural transformation, as a binary switch activated by cellular signals, is a general mechanism in gene regulation networks [36]. We further investigated the effects of RDDs on RNA secondary structure and found that 171 RDDs in 117 genes can lead to the changes of their RNA secondary structure (Appendix A). Among them, the minimum free energy (MFE) of 61 genes increased after RDD occurred, while that of the other remaining 56 sites decreased (Appendix A). As we know, MFE can be used to measure the stability of the RNA molecule, and a structure with a low MFE value is more stable [37,38]. According to our prediction, drought responsive RDD could reduce the stability of 61 genes and also increase the stability of the other 56 genes. The secondary structure changes of these genes will impact on achieving their function, indicating that the RDDs might be involved in regulating drought response through influencing the structural stability of the drought-related genes. Further study on the specific role of RDD in regulating RNA secondary structure will contribute to better understanding of the genetic basis of crop drought resistance.

miRNA is a class of small non-coding RNAs that often bind to complementary sequences of the mRNA targets to silence or weaken their expression [30]. RDD generally caused variations in mRNA sequence, which could make the transcripts gain or lose miRNA binding [39]. To understand the effects of drought responsive RDDs on miRNA targeting, we further investigated miRNA-mRNA targeting pairs using the genes with or without RDD mutations as target genes. Results showed that there were 160 genes that were potentially targeted by 81 unique high-confidence mature miRNAs of wheat, containing 236 RDDs (Appendix A), of which 3 RDDs occurring in 3 genes were detected to result in the changing of miRNA targets compared to the initial genes without RDD. We found two RDD sites (chr2A_21252947: A to T; chr2A_21252961: A to C) occurring in TraesCS2A02G053100 (Figure 5A): the T to A site changed the amino acid from Phe to Tyr (Figure 5B), and the T to G site made it lose the binding of tae-miR9657b-5p (Figure 5C). Meanwhile, the RNA secondary structure of this gene was also changed by the RDD sites (Figure 5D), that the MFE value changed from −657.40 to −659.40 kcal/mol. Furthermore, the difference efficiency of T to G in TraesCS2A02G053100 is 0.18 under the drought condition while not occurring in CK with the variation efficiency of 0 (Figure 5E). The expression level of this gene was also slightly higher under the drought condition compared to CK (Figure 5F). These results indicated that RDD could precisely regulate the expression level of the target genes through mediating miRNA targeting with variable variation efficiency. In addition, orthologs analysis found that TraesCS2A02G053100 is the ortholog of *OsRLCK*. *OsRLCK* is reported as the key candidate associated with QTL for drought stress related traits [40], suggesting that TraesCS2A02G053100 could also have potential function in response to drought stress. The RDD occurring in this gene might play an indispensable role in regulating the sophisticated process of drought response that this gene was involved in.

### 2.5. Effects of RDDs on Protein Conserved Domain

RNA/DNA difference is a post-transcriptional modification mechanism to increase proteomic diversity by modifying protein composition, protein structure and binding ability [41]. To understand the effect of drought-responsive RDD on the encoded protein structure, we investigated and compared the conserved domain organization of the genes with missense variant and stop gained due to RDD before and after RDD occurred. The results showed that 2 genes were impacted by RDDs to change their protein domain organization (Appendix A), suggesting that RDDs might have effects on protein functions. One is TraesCS6D02G181700, which is the orthology of *AtABA2*, having two RDD sites (chr6D_208258669; chr6D_208258671) and site chr6D_208258671, changing the amino acid from Val to Ala to provide an wpimerase domain after RDD occurred. It is reported that AtABA2 encoded xanthin dehydrogenase, which is a key enzyme for abscisic acid (ABA) biosynthesis, to play the crucial role in regulating ABA levels in Arabidopsis [42,43]. The other gene is TraesCS7A02G162400, which is the orthology of G-protein coupled receptor 1 (*GCR1*) in Arabidopsis. A RDD site T to C (chr7A_118456566) in TraesCS7A02G162400 resulted in loss of the transmembrane receptor domain although the slime mold cyclic AMP receptor domain was present (Figure 6A,B). Previous studies have demonstrated that *GCR1* was an important regulator involved in stress signal transduction, such as drought stress, ABA response and regulation of stomatal aperture in Arabidopsis [44,45]. Furthermore, we found that the expression level of this gene was also down-regulated after RDD occurred (Figure 6C), and the protein three-dimensional (3D) structure predictions also displayed differences before and after RDD (Figure 6D). These results indicated the drought-responsive RDD could also impact on the protein domain of the drought-related genes and act as the regulator in the biological process of drought response and tolerance.

## 3. Discussion

Plants have evolved diverse regulatory mechanisms to deal with environmental stresses at both the transcriptional and post-transcriptional levels [46,47]. From the perspective of post-transcriptional regulation, little is known about the role and function of RNA/DNA difference in response to drought although some studies have been conducted to identify alternative splicing events associated with drought stress [48,49]. Here, we conducted the first global survey of drought-related RDDs in wheat and 265 drought-responsive RDDs in 176 genes were obtained. Furthermore, we validated one drought-responsive RDD using the RT-PCR combined with Sanger sequencing. This study not only proved that RDD events actually occurred in the nuclear genome in wheat, but also demonstrated that they were involved in regulating drought response. Although we cannot assure that all of the remaining 264 RDD sites were in actual existence due to possible false positive results, we used the most rigorous and reliable current approaches to obtain reliable results. Firstly, the RNA-Seq dataset from the same accession used for generating the reference genome was adopted to identify RDDs, which completely ruled out the interference of genotype-specific mutation and polymorphism. Secondly, three independent biological replications were adopted and only the RDD found in all of the 3 replications was retained, which could excluded sequencing errors or other random mistakes. Thirdly, comparison of the control and drought- stressed sample combined with manual verification through IGV checking were used to obtain the high-confidence drought-responsive RDDs, which can remove the mapping errors. At the same time, the wheat genome is one of the most complex genomes in plants, with A, B and D subgenomes as well as more than 80% repetitive elements, which could result in false positive mapping when RNA-Seq is mapped on the reference genome causing false positive RDD results [50]. Thus, the identified drought-responsive RDD reported here were reliable within the current level of knowledge and technology.

Furthermore, we found there is no significant correlation of RDD with its gene expression. Only a small part of genes with drought-responsive RDDs displayed differential expression under drought stress compared to the control, indicating that RDD exerted slight influence on gene expression. Then, we detected the effects of RDDs on the RNA secondary structure. Out of 265 drought-responsive RDDs, 171 RDDs lead to changes of RNA secondary structure in 117 RDD-containing genes, of which 61 genes reduced the stability and 56 genes increased the stability due to RDDs, suggesting that RDDs impacted significantly on the RNA composition and then affected the RNA structure. It is reported that post-transcriptional gene expression mediated by small non-coding RNAs (such as microRNAs) and RDD events regulatory processes are connected at a fundamental level [51]. In humans, the functional relationship between RDDs and miRNA mediated post transcriptional gene silencing has been revealed [51]. RDD can result in the gain or loss of miRNA target sites in the RDD-containing genes [30]. We further predicted and compared the miRNA targets of all RDD-containing genes affected by RDDs, and found that 3 RDD events caused the change of miRNA targeting. In particular, T to G difference in TraesCS2A02G053100 caused it to lose the binding of tae-miR9657b-5p, and then increase its expression under the drought condition. Orthologous gene analysis found that TraesCS2A02G053100 might be a drought-related gene. It is noticed that RDD occurs alongside efficiency, suggesting that it can regulate the expression level of RDD-containing gene with a very precise method; this may quantify the difference efficiency to produce the quantitative transcript, and then mediate the miRNA to participate in the drought response process. Further study of the role of RDD in mediating miRNA targeting could elucidatea novel regulatory mechanism or network underlying drought response and tolerance in wheat [51].

Generally, one important effect of RDDs is to increase the diversity of the proteome. Here, 75 out 265 drought-responsive RDDs were found to cause the missense variants, which contributed to enriching proteomic diversity. The effects of RDD on the protein conserved domain and 3D structures were also investigated. As we know, structure is the basis of protein function. The change of protein domain and 3D structure caused by RDDs might also affect the protein’s function. In this study, we found that a droughtresponsive RDD (T to C) in TraesCS7A02G162400, which is the orthology of *AtGCR1*, resulted in loss of the transmembrane receptor domain and also changed its 3D structure. It is reported that *GCR1* acts as a negative regulator of GPA1-mediated ABA response in Arabidopsis guard cells, and loss-of-function mutants of *AtGCR1* showed resistance to drought stress and activated the high expression of some known drought and ABA regulatory genes under drought stress [45]. The drought-responsive RDD in TraesCS7A02G162400 might regulate this gene to be involved in the drought stress response through functions on the protein domain and 3D structure. Further functional studies of this RDD are needed to reveal the its role in regulating drought stress, enriching the epigenetic mechanism of drought response and tolerance in wheat.

## 4. Materials and Methods

### 4.1. RNA-Seq Data and Mapping

The RNA-Seq dataset used in this study was downloaded from Sequence Read Archive database with the accession no. SRP098756 [52]. Raw data was quality controlled by FastQC (version 0.11.8) and Trimomatic (version 0.39) to remove adapter reads, low-quality reads, or contamination. The obtained high-quality reads were aligned to wheat reference genome (IWGSC RefSeq version 1.1) using STAR tool (version 2.7.6). The alignment file was subsequently analyzed using StringTie (version 1.3.5) and subsequently the gtf file was combined using the merge function in StringTie of each sample. In addition, we quantified the read coverage of each gene by featurecount (version 0.11.2). Differential gene expression analysis was performed by using DESeq2 tool with the adjusted P value as less than 0.05 and |log2FC| > 1.

### 4.2. Identification of RNA/DNA Difference Sites

These RNA-Seq reads were aligned to the wheat reference genome and removed from the same location reads using MarkDuplicates tool in Picard [53]. Then, the SplitNCigarReads tool in GATK (Genome Analysis Toolkit) software was used to separate the reads on the exons, remove the N error bases, and remove the reads in the intron region. Then, the AddOrReplaceReadGroups tool in GATK software was used to assign all read operations in a file to a new read group. Finally, the HaplotypeCaller tool in GATK software was used to call DNA/RNA differences. Subsequently, DNA/RNA differences of each sample were obtained and stored in the raw VCF file for further analysis to obtain the potential RDDs as follows:(1)GATK VariantFiltration tool was used to correct the systematic errors of the sequencing platform and software, and the initial filtering parameters – Filter “FS > 30.0”, – Filter “QD < 2.0” were selected;(2)The site was mapped by more than 10 reads with reference read >2 and mutation read >3 retained;(3)In order to improve the accuracy, only the DNA/RNA difference sites that appeared in all of the three replicates were retained;(4)the RDDs between control and drought-treated samples were compared to remove the same sites to obtain the potential drought-responsive RDDs;(5)manually verify each bam file to assure the RDD’s actual occurrence through Integrative Genomics Viewer (IGV) tool.

Through these filtering, the high-confidence RDDs associated with drought were finally obtained. Then, we used the SnpEff tool (version 3.6) and the annotation file downloaded from the Ensembleplants database [54] to annotate the filtered VCF file.

### 4.3. Validation of RNA/DNA Difference Site Using RT-PCR

For experiment verification, the seedlings of genotype CS42 grown under normal and drought (19.2% PEG) conditions were collected to isolate RNA using the RNA Easy Fast Plant Tissue Kit (Tiangen, Beijing, China). Then, total RNA was used to synthesize cDNAs using RT Master Mix Perfect Real-Time kit (Takara, Dalian, China) according to the manufacturer’s instruction. We randomly selected one RDD (T to C) in TraesCS3A02G045300 for validation by RT-PCR analysis using the forward primer and reverse primer of GAAGATCTGGTTTCCATGGGACT and GCAAATCATGCAGTGGAATCAAGC, respectively.

### 4.4. RNA Secondary Structure and MiRna Target Analysis

The RNA secondary structure of the RDD-containing gene was predicted by the RNAfold online tool [55] using the default parameters. To determine whether RDDs affected miRNA targeting, all of the RDD-containing genes were searched against the publish 119 wheat miRNAs in the miRBase database using psRNATarget tools [56]. Schema V2 (2017 release) model was used to score the possibility of miRNA targeting on the gene, and the result with the lowest expectation was selected as the optimal prediction. Then, the effects of RDDs on miRNA binding were obtained through comparison of the prediction results of whether RDD occurred or not.

### 4.5. Conserved Domain Prediction and Protein Structure Analysis

The PFAM database (version 33.0) was used to search protein domains based on HMMER3 (version v3.1.1) for RDD-containing genes using protein sequences as input with the E-value < 1 × 10^−5^. The 3D structure was predicted using the homologous modeling method in SWISS-MODEL database [57]. The model with the highest degree of coincidence with the target protein and more than 30% was adopted.

## 5. Conclusions

Here, we identified the first drought-responsive RDD landscape in wheat, proving that RDD events actually occurred in the wheat nuclear genome and were also involved in regulating drought response. Furthermore, the effects of these drought-responsive RDDs were systematically investigated, showing that these RDDs could exert diverse effects on the drought-related genes to mediate drought response and tolerance. They could impact on the expression, RNA secondary structure, miRNA targeting as well as protein domains and 3D structure of the RDD-containing genes. The identified drought-responsive RDDs and their RDD-containing genes provide important resources for mining new genes associated with drought stress. This study lays the foundation for further functional studies to better understand the epigenetic mechanisms underlying drought stress, and also pave the way to identify RDDs in the nuclear genome based on RNA-Seq data in wheat and beyond.

## Figures and Tables

**Figure 1 ijms-23-01405-f001:**
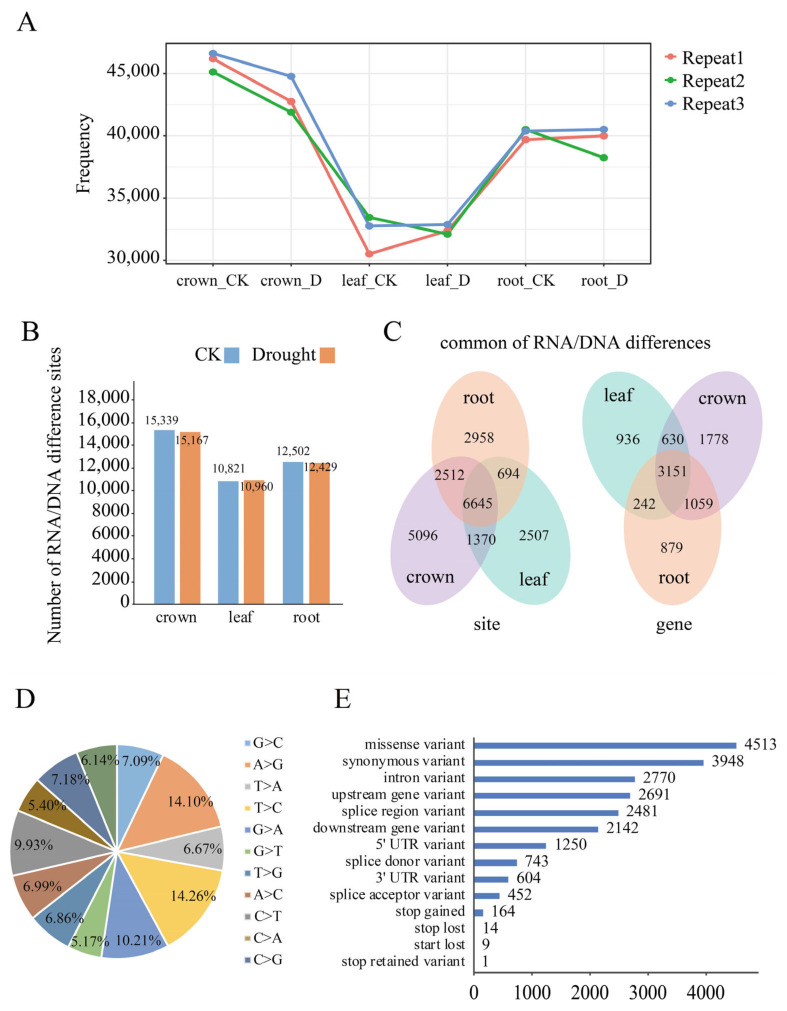
Characterization of RNA/DNA differences in wheat nuclear genome using RNA-Seq data. (**A**) The number of variations identified in three biological replicates of control and drought conditions in the crown, leaf and root, respectively; (**B**) The numbers of unique RNA/DNA differences identified in crown, leaf and root, respectively; (**C**) The shared and unique RDD sites (Left) and RDD-containing genes (Right) found in crown, leaf and root tissues; (**D**) The variation types of the identified unique RDDs; (**E**) The frequency distribution of RDDs in the transcription regions. The *y*-axis represents different types of gene regions, and the *x*-axis represents the abundance of RDDs.

**Figure 2 ijms-23-01405-f002:**
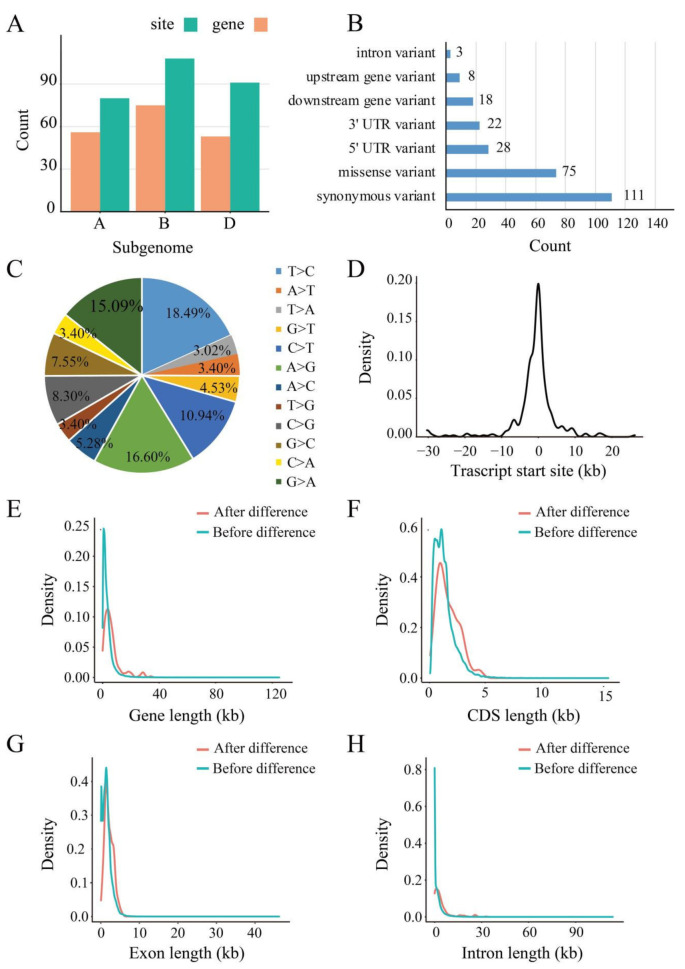
Characterization of the drought-responsive RDDs. (**A**) The numbers of drought-responsive RDD and related-genes in A, B, D subgenome, respectively; (**B**) The distribution of drought-responsive RDDs in the transcription regions. The *y*-axis represents different types of regions, and the *x*-axis represents the abundance of RNA/DNA difference sites; (**C**) The variation types of the identified drought-responsive RDDs; (**D**) The distance of the distribution of drought-responsive RDD site and the TSS (transcription start site) in the RDD-containing genes; (**E**–**H**) Comparison of gene length, CDS length, exon length, and intron length of gene without RDD and with RDD.

**Figure 3 ijms-23-01405-f003:**
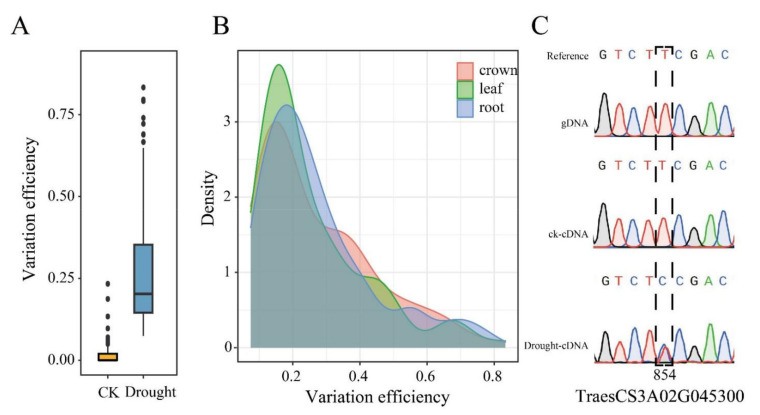
Variation efficiency and validation of these identified RDDs. (**A**) Comparison of the variation efficiency of RDDs between CK and drought conditions; (**B**) the distribution of variation efficiency in 3 different tissues; (**C**) Validation of the randomly selected drought-responsive RDD in TraesCS3A02G045300 through RT-PCR combined with Sanger sequencing.

**Figure 4 ijms-23-01405-f004:**
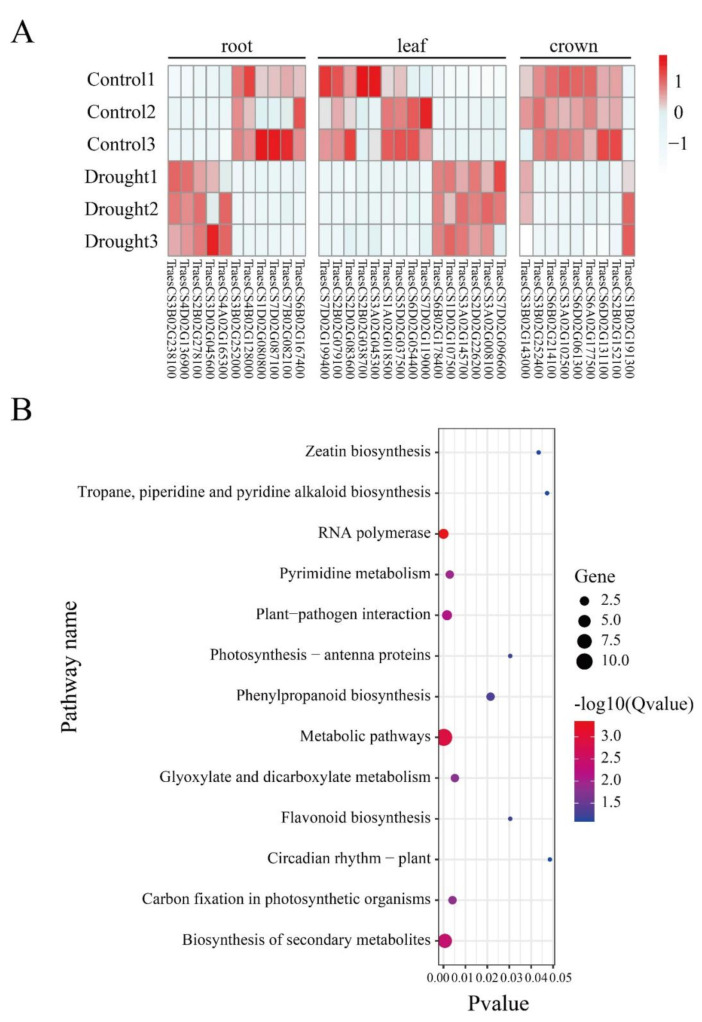
Differential expression and functional enrichment of the genes occurring in drought-responsive RDD. (**A**) The differential expression of RDD-containing genes between CK and drought conditions in roots, leaf and crown tissues; (**B**) KEGG enrichment analysis of these differentially expressed RDD-containing genes.

**Figure 5 ijms-23-01405-f005:**
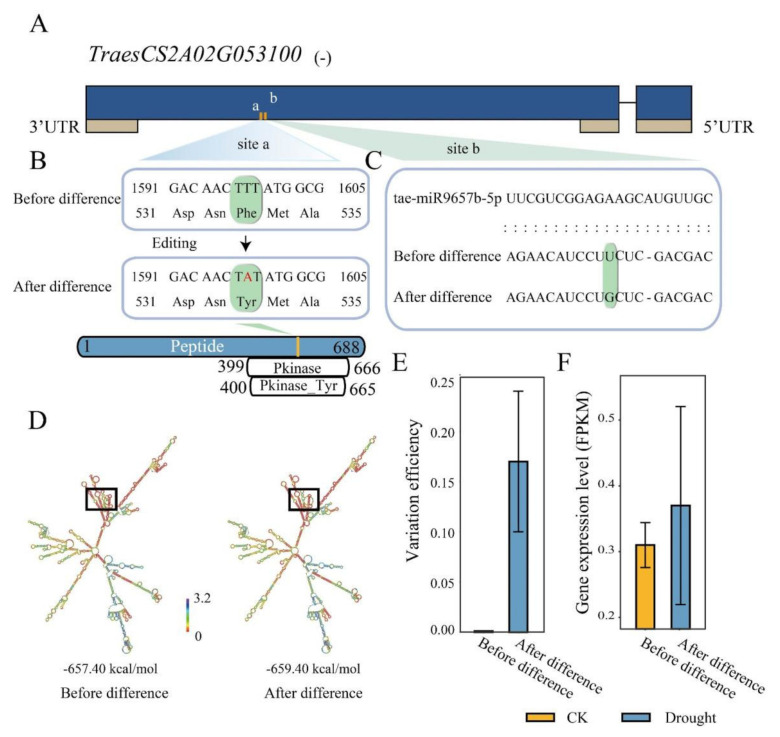
RNA/DNA difference functions on the miRNA targeting in TraesCS2A02G053100. (**A**) Two RNA/DNA difference sites (a, b) were found in TraesCS2A02G053100, which is annotated in the reverse chain of the genome; (**B**) The RDD site A to T (site a) caused amino acid Phe changed into Tyr; (**C**) The RDD site A to C (site b) leads to the loss of miR9657-5p targeting; (**D**) Change of the RNA 2D structure before and after RDD variations; (**E**) Variation efficiency of the A to C site (site b); (**F**) Expression levels of TraesCS2A02G053100 before and after variation.

**Figure 6 ijms-23-01405-f006:**
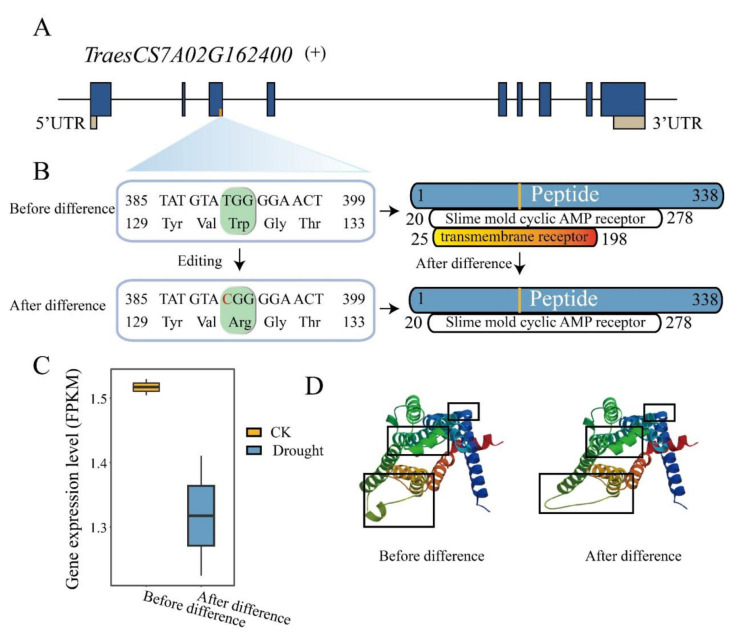
RNA/DNA difference impacts on the protein conserved domains and 3D structure in TraesCS7A02G162400. (**A**,**B**) A T to C variation was identified in the third exon of TraesCS7A02G162400, which caused Trp to Arg change, and lost a protein conserved domain; (**C**) Expression levels of TraesCS7A02G162400 before and after variation; (**D**) Changes of protein 3D structure before and after difference.

## Data Availability

The data that supports the findings of this study are available in the Appendix A of this article.

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
