# Peer review of "Genome-Wide Identification and Characterization of RNA/DNA Differences Associated with Drought Response in Wheat"

_ijms, 2022, doi:10.3390/ijms23031405_

Round 1

Reviewer 1 Report

The article entitled "Genome-wide identification and characterization of RNA/DNA differences associating with drought response in wheat" is well planne, excuted and written.However it requires minor changes.

Minor comments
Q1 Why authors used IWGSC RefSeq version 1.1 under, line 347 "4.1RNA-seq data and mapping" while the updated version is available (IWGSC RefSeq v2.1 Assembly)

Q2 Put a different subheading line"cDNA synthesis and RT-PCR" or any other  for line starting from line 377 to 384

Q3 Authors should minimise the unwanted abbreviation such as "conserved extended domain (E)", "C-terminal (PPR-E)",  "ADAR (adenosine deaminase acting on RNA) " pentatripeptide repeat (PPR)"  it create confusion.

Q4 Line 28: Triticum aestivum L. should be italic
Q5 line 50 Authors wrote "RDD (RNA editing)" RDD means RNA ediing or RNA/DNA differences (RDDs) at line 40 
Q6 Line 216 remove " (see Methods for more details"
Q7 Line 354: remove "firstly"
Q8 Line 386: The word "RDD gene" not seems to be coorect as RDD is process. Write the appropriate word. (correct at other place also in manuscript).
Q9 Line 395: What does it mean "(33.0 release)" ? correct it. 

Author Response

Comments and Suggestions for Authors

Reviewer 1

Question1: Why authors used IWGSC RefSeq version 1.1 under, line 347 "4.1RNA-seq data and mapping" while the updated version is available (IWGSC RefSeq v2.1 Assembly)

Response: We appreciated the reviewer’s insightful and constructive comment. We agreed that it is better to use the updated version IWGSC RefSeq v2.1 (Zhu et al., 2021) as reference to perform analysis. But when we started to conduct this study in March 2021, the annotation information of the IWGSC RefSeq v2.1 is not publicly available. Thus, we had to use the IWGSC RefSeq version 1.1 as the reference. Although the obtained RDD sites might show some differences between the two reference version, the patterns of RDD and the identified drought-responsive RDDs  could remain the same. As we known, compared to version 1.1, version 2.1 significantly increases the the numbers of transposable elements (TEs) and intact TEs in wheat reference as well as corrects the contig positions and orientations. The high-confidence (HC) genes are not obviously different between them with version 1.1 of 105,319 and version 2.1 of 105,534 (Zhu et al., 2021). As we known, the reference is just used as template and then the RNA-seq reads of CK and drought stressed samples were mapped on the reference to identify RDD. Therefore, the different version would not have significant impact on the results.

Question2: Put a different subheading line"cDNA synthesis and RT-PCR" or any other for line starting from line 377 to 384

Response: We appreciated the reviewer’s constructive suggestion. We have added the subhead “Validation of RNA/DNA difference site using RT-PCR” as suggested.

Question3: Authors should minimise the unwanted abbreviation such as "conserved extended domain (E)", "C-terminal (PPR-E)",  "ADAR (adenosine deaminase acting on RNA) " pentatripeptide repeat (PPR)"  it create confusion.

Response: We are grateful to the reviewer for pointing out this problem. We have carefully revised the abbreviation throughout manuscript to avoid any confusion.

Question4: Line 28: Triticum aestivum L. should be italic

Response: We have fixed it.

Question5: line 50 Authors wrote "RDD (RNA editing)" RDD means RNA ediing or RNA/DNA differences (RDDs) at line 40 

Response: We appreciated the reviewer’s insightful comment. RDD means RNA/DNA difference. We have revised it to avoid any confusion.

Question6: Line 216 remove " (see Methods for more details"

Response: We have removed it as suggested.

Question7: Line 354: remove "firstly"

Response: We have deleted it as suggested.

Question8: Line 386: The word "RDD gene" not seems to be correct as RDD is process. Write the appropriate word. (correct at other place also in manuscript).

Response: We appreciated the reviewer’s insightful and constructive comment. We have revised "RDD gene" to " RDD-containing gene" throughout the manuscript.

Question9: Line 395: What does it mean "(33.0 release)" ? correct it. 

Response: We are grateful to the reviewer for pointing out this problem. We have changed "(33.0 release)" to "(version 33.0)"

Reviewer 2 Report

The manuscript is well written with a lot of information. The RNA/DNA difference in abiotic stress is a newly emerging area and is highly recommended for publication. The abstract needs to be modified with less technical language so that readers from not the same field, also can develop an interest in this work. Rest is fine.

Author Response

Comments and Suggestions for Authors

Reviewer 2

The manuscript is well written with a lot of information. The RNA/DNA difference in abiotic stress is a newly emerging area and is highly recommended for publication. The abstract needs to be modified with less technical language so that readers from not the same field, also can develop an interest in this work. Rest is fine.

Response: We appreciated the reviewer’s professional and constructive comment. We have polished the abstract to make it understandable and readable.